# Improving RAG Question Answering Generation with Synthetic Data

## Abstract

To improve large language models (LLMs) for question answering (QA) tasks, system architects often look to retrieval-augmented generation (RAG) or fine-tuning approaches to increase a model's performance. In many applications, however, there is a dearth of real data of sufficient quality to support model fine-tuning to improve RAG system performance for QA tasks. In this work, we study the degree to which synthetic data can effectively substitute real data for RAG generator fine-tuning. Using GPT-4o, we generate a synthetic version of the HotpotQA training set and fine-tune a Llama-3 generator separately on both real and synthetic data. We evaluate our models with a range of metrics such as token-level F1, Bertscore, and LLM-as-a-judge. Across these metrics, model performance generally increases after fine-tuning primarily due to better conformity to the style of the answer distribution and secondarily due to improved use of retrieved contexts. We observe that relative performance depends on the quality of the retriever, emphasizing the importance of the training data distribution in improving the model's reasoning over multiple contexts. We further show that the fine-tuned model trained on synthetic data generalizes better to similar held-out QA tasks, outperforming an LLM fine-tuned on real data by 36% in LLM-judged correctness over the RepLiQA dataset. These findings motivate a system-level analysis of the marginal benefits of generator fine-tuning in RAG pipelines, providing practical insights on the utility of synthetic training data for the benefit of both RAG systems engineers and future researchers.

## 1 Introduction

*Question answering* (QA) is the task of providing machine-generated answers to a user query and is useful in a variety of domains, including medical, finance, and cybersecurity (Singhal et al. (2025); Tao et al. (2024); Agrawal et al. (2024)). The advent of *retrieval-augmented generation* (RAG) (Lewis et al., 2020) has rapidly increased the performance of large language models (LLMs) on a variety of QA benchmarks and tasks (Fan et al., 2024). RAG systems propose to, given a large corpus of documents, retrieve and inject relevant context into the LLM generator to support the system in answering correctly. These systems have been especially useful for tasks in specialized domains or where additional documents are needed beyond what can fit in an LLM's context window.

Past research has proposed a variety of methods to boost RAG outcomes, including fine-tuning the generator (Liu et al. (2025b); Zhang et al. (2024)), retriever (Karpukhin et al. (2020); Gao & Callan (2022)), or both (Siriwardhana et al. (2023); Mao et al. (2024)). Although prior studies suggest retriever fine-tuning will have the greatest impact on wholistic system performance in many contexts, generator fine-tuning has been shown to improve results (Zhang et al., 2024).

Limits on computational resources, however, may make frequent training of the generator impractical, particularly if the same generator is applied across several tasks, the model size is large, or the underlying data distribution continually evolves. Furthermore, many applications lack sufficient training data to make an impact. This may be due to a variety of reasons:

1. A scarcity of real data, especially labeled QA examples.

2. Inability to fine-tune on real data due to sensitivity, privacy, or other concerns.

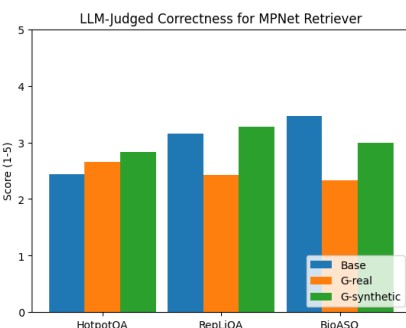

Figure 1: Summary of experimental setup. We take two copies of a Llama-3 (Grattafiori et al., 2024) LLM and fine-tune on real and synthetic versions of the HotpotQA train set, producing *G-real* and *G-synthetic*, respectively. We then evaluate each on the in-domain HotpotQA dev set along with both RepLiQA and BioASQ.

Figure 2: LLM-judged correctness for each LLM under the MPNet Retriever. 1) *G-synthetic* outperforms *G-real* in all cases. 2) *G-synthetic* generalizes to a held-out in-domain task, RepLiQA, better than the baseline *G-real*, although the baseline wins on an out-of-domain task.

In this work, we study the fine-tuning of a RAG generator on synthetic data to improve performance on QA tasks. We fine-tune a Llama-3 (Grattafiori et al., 2024) based model on the HotpotQA question answering dataset (Yang et al., 2018). Additionally, we develop a protocol to generate a dataset from the same HotpotQA contexts and fine-tune a second Llama-3 based LLM on this synthetic QA data. We evaluate each LLM based on both objective (exact match and F1) and subjective (LLM-as-a-judge) metrics under two retriever setups using QA pairs from both HotpotQA, RepLiQA (Monteiro et al., 2024), and BioASQ (Krithara et al., 2023). Our overall experimental setup is summarized in Figure 1.

Our contributions are as follows:

1. Demonstrate that a synthetically fine-tuned generator can not only match, but under an imperfect retriever even improve RAG performance over real data (see Figure 2).

2. Show that a synthetically tuned LLM exhibits superior generalization performance to its real counterpart across all metrics.

3. Provide an LLM-as-a-judge analysis which shows that higher exact match, F1, and correctness scores result primarily from the LLM improving in generating answers according to the style of the target distribution and secondarily by improved reasoning over the contexts themselves.

4. Observe that the ideal fine-tuning distribution depends on the accuracy of the retriever.

We publish our synthetically generated dataset as well as a lightweight package wrapping LlamaIndex[1] to create simple RAG engines.

---
[1]https://www.llamaindex.ai/

| Prior Work | Target task | Approach |
|---|---|---|
| RAG-end2end (Siriwardhana et al., 2023) | Domain-specific ODQA | Joint training of retriever and generator |
| ChatQA (Liu et al., 2025b) | Conversational QA | New benchmark; multi-stage fine-tuning process |
| RAFT (Zhang et al., 2024) | Open-book QA | Chain-of-thought; ignore distractor documents |
| RAG-Studio (Mao et al., 2024) | Domain-specific QA | Jointly train retriever and generator on in-domain synthetic data |

Table 1: A summary of related work on RAG fine-tuning for QA tasks.

We proceed as follows: We begin by summarizing related work. We then detail our experimental setup and summarize our results. We conclude with a discussion of the results and avenues for future work.

## 2 RELATED WORK

When adapting LLMs to a target task, practitioners often choose between fine-tuning and RAG (Balaguer et al., 2024). We summarize related work on each approach, analyze contributions at the intersection of these two methods, and finally cite applications of synthetic data cutting across each of these methodologies.

### 2.1 FINE-TUNING

Fine-tuning may be performed as continual pre-training, domain-adaptive pre-training, or continual fine-tuning (Shi et al., 2025). *Continual pre-training* seeks domain shift across language, content, or time, while *domain-adaptive pre-training* aims to adapt to a new target domain entirely. Although both approaches can be effective (Qin et al. (2022); Shen et al. (2024)), computational, data sensitivity, or other limits may make them impractical in many use cases. For this reason, we study *continual fine-tuning*, a less data-intensive adaptation of the LLM to a target task—in our case, QA (subsequently, we refer to this process simply as 'fine-tuning').

Fine-tuning may be performed as full-weight fine-tuning or adapter weight tuning, such as LoRA (Hu et al., 2021). Recent work suggests LoRA performs worse on out-of-domain tasks (Shuttleworth et al., 2025). Since we aim for our generalization across domains or tasks, exclusively analyze full-weight fine-tuning for this work.

### 2.2 RETRIEVAL-AUGMENTED GENERATION

Retrieval-augmented generation (RAG) (Lewis et al., 2020) consists of two components: a retriever and a generator. The retriever $p_\eta(z|x)$ pulls the top $k$ documents relating to a user input $x$; the generator $g_\theta(y_i|x, z, y_{0:i-1})$ uses these documents, along with the user input and previous content, to predict the next-most likely token. RAG systems are useful when the knowledge corpus is larger than the LLM context window and can be applied to a variety of open- and closed-domain QA tasks (Siriwardhana et al. (2023); Opoku et al. (2025)). Several works have sought to improve performance in this area through techniques such as incorporating relevance estimators for the contexts (Kim & Lee (2024); Wang et al. (2024)), improving retrieval embeddings (Gao & Callan, 2022), and engaging the generator $g_\theta$ in the retrieval process (Asai et al., 2023).

### 2.3 RAG + FINE-TUNING

A growing body of literature has explored the combination of fine-tuning with RAG. We summarize their contributions in Table 1. Unlike other works which primarily focus on overall system performance based on F1 score, we study the specific impact of fine-tuning on synthetic data on answer correctness, applying an LLM-as-a-judge model.

## 2.4 SYNTHETIC DATA

The use of synthetic data for RAG systems has been studied for large dataset generation (Dai et al., 2022) and performance improvement (Liu et al. (2025b); Zhang et al. (2024); Mao et al. (2024)). Notably, although Liu et al. (2025b) performs an ablation study demonstrating that fine-tuning real and synthetic data achieves near-equivalent performance, our comparison across retriever quality and LLM-as-a-judge analysis are both novel. In addition, our work varies from Zhang et al. (2024) in that we focus our analysis on the comparison of results with synthetic and real data.

## 3 EXPERIMENTAL SETUP

Our overall approach is as follows: first, we identify a generalized multi-hop QA dataset, HotpotQA, as our target task. We fine-tune two separate versions of an LLM, Llama-3.1 8B Instruct,[2] for this RAG task, one with HotpotQA data and the other with synthetically generated multi-hop QA pairs. We call these fine-tuned LLMs *G-real* and *G-synthetic*, respectively. We place each of these models, along with the *base model* without fine-tuning, inside a RAG context. We evaluate the results of each through a combination of objective and LLM-as-a-judge metrics. Our experimental setup is detailed in Figure 1.

## 3.1 DATASETS

We identify HotpotQA as our primary dataset for training and evaluation. HotpotQA has been widely applied (Zhang et al. (2024); Dettmers et al. (2023)) as a benchmark for the open-domain question answering task we study here. HotpotQA consists of a train set of 90,447 examples and dev and test sets of 7,405 examples each.[3] Each question contains two *golden contexts*, or passages, which must be used in tandem to answer the question. We selected this dataset for this multi-hop characteristic and broad, open-domain knowledge (Wikipedia articles). This is important for our study as multi-hop questions are more realistic for many use cases in requiring knowledge synthesis from a variety of source documents.

To test the generalization of *G-real* and *G-synthetic* on new tasks, we select RepLiQA and BioASQ as alternative evaluation datasets. RepLiQA contains questions previously unseen by LLMs, meaning they must rely on a proper reading of the retrieved context to answer the question. The answers are of a similar traditional open-domain QA style as those of HotpotQA. To test generalization to specialized domain knowledge, we select BioASQ, a dataset of multi-hop biomedical QA pairs. Because of the specificity of the domain and the unique answer style, we expect the fine-tuned LLMs to perform worse than the baseline on BioASQ.

### 3.1.1 SYNTHETIC DATA GENERATION

To generate a synthetic benchmark dataset, we begin with the same golden passage pairs as the HotpotQA train set. Specifically, we are given 90,447 pairs of paragraphs from Wikipedia pages which hyperlink to each other. We use a custom prompt to ask GPT-4o (gpt, 2024) to generate multi-hop questions and corresponding answers for each of the given text passage pairs. GPT-4o is instructed to use only the provided context(s) to generate the question-answer pair and to avoid using pre-trained knowledge. Our synthetic data generation prompt is presented in Listing 2.

To characterize the differences between the real and synthetic datasets, we compare the embedding cosine similarity between the questions and their relevant (golden) contexts in Table 2. We observe that the synthetically generated questions exhibit higher similarity to their relevant contexts, suggesting they are better grounded in the passages. Notably, we observe that this holds for both the least and most similar contexts to each question, indicating the synthetic questions remain sufficiently 'multi-hop.' In addition to this quantitative analysis, we provide qualitative comparisons in Figure 3. We note that:

---

[2]https://huggingface.co/meta-llama/Llama-3.1-8B-Instruct

[3]A fullwiki version of the dev and test set using passages retrieved from their own basic retriever is available without the golden contexts; however, we elect to use our own retriever. For all experiments, we use v1.1 of the dataset, available here: https://hotpotqa.github.io/.

| Dataset | Min | Max | Harmonic Mean |
|---------|-----|-----|---------------|
| Real | 0.461 | 0.641 | 0.536 |
| Synthetic | **0.476** | **0.652** | **0.550** |

Table 2: MPNet embedding cosine similarities between real and synthetic HotpotQA questions and their relevant (golden) contexts. We report the average min (least similar context), max (most similar context) and the harmonic mean between the two.

1. Both real and synthetic questions are prone to not adequately leveraging both contexts to create a multi-hop question.

2. The synthetic questions sampled can be more complex and of better style.

A key point here is the usage of the same passage pairs as in the real training set, which allows us to perform an apples-to-apples comparison between real and synthetically generated questions. Because of this, we do not contribute any additional methods for generating multi-hop candidate passage pairs to the literature; however, the HotpotQA paper and others have proposed methodologies for this task.

## 3.2 RAG SYSTEM SETUP

Our RAG system consists of two components: the retriever and generator. For the retriever vector store, we rely on the all-mpnet-base-v2 embedding model (Song et al., 2020).[4] We construct the same ChromaDB index containing all HotpotQA dev set passages and use this for all experiments.[5] For the MPNet retriever, we pull the top $k = 3$ contexts for each query. We perform no fine-tuning on the MPNet retriever as our goal is to assess the performance increase from generator fine-tuning only.

As a baseline, we implement a GoldenRetriever which returns the golden passages for each query. A description of the GoldenRetriever may be found in Algorithm 1. The GoldenRetriever by design only returns the golden passages relevant to each query. Note that this means our experiments with MPNet necessarily involve distractor documents (both from the extra document and missed retrievals), whereas the GoldenRetriever experiments have no distractors present.

We measure the success of the retriever via hit rate and mean reciprocal rank (MRR). Hit rate measures the proportion of queries which retrieved at least one relevant document. MRR is defined as

$$MRR = \sum_{i=1}^{|R|} \frac{1}{rank_i} \qquad (1)$$

where $R$ is the set of retrieval queries and $rank_i$ is the 1-indexed rank of the first golden context retrieved for retrieval $i$. You can find the results for both retrievers in Table 8. We note that the MPNet retriever is a significant limitation to the generator as opposed to the GoldenRetriever; for HotpotQA, 27% of the queries turn up no relevant passages whatsoever, and likely much fewer have both golden contexts. Trivially, the GoldenRetriever achieves perfect hit rate and MRR since it always returns the relevant contexts to queries present in the HotpotQA dataset.

## 3.3 FINE-TUNING

We use Llama 3.1-8B Instruct for all experiments (Grattafiori et al., 2024). Consistent with prior work, we only compute the loss over the LLM's answer to the RAG system query and not the question itself. This means the LLM is only optimized to generate the correct answer and not memorize the passages themselves. We do this for two reasons:

---

[4]https://huggingface.co/sentence-transformers/all-mpnet-base-v2

[5]https://www.trychroma.com/

| Hyperparameter | Value |
|---|---|
| Gradient Accumulation Steps | 2 |
| Micro Batch Size | 4 |
| Epochs | 1 |
| Learing Rate Schedule | Cosine |
| Learning Rate | 2e-5 |
| Warmup Steps | 100 |

Table 3: Fine-tuning hyperparameters.

1. To prevent leakage of the HotpotQA passages into the LLM, skewing the results. There is overlap between the HotpotQA train and dev set passages.

2. To support our use case. Since we aim for a generator that can be applied to an evolving knowledge domain or out of-distribution data, it should not rely on parametric knowledge for the response.

We also do not train on *distractor*, or irrelevant, documents as our goal is to primarily compare the impact of fine-tuning on real versus synthetic data. Zhang et al. (2024) note better results when introducing distractor documents, so we leave the work of integrating this with our work as future research. We note that our evaluation, however, does necessarily involve distractor documents due to imperfections in the MPNet retriever.

Our RAG system prompt, used for both training and evaluation, is provided in Listing 1. We use the axolotl library for fine-tuning (axo, 2025) along with Deepspeed ZeRO 3 (Rajbhandari et al., 2020) and 16 floating point weights for efficiency. Our parameters for fine-tuning may be found in Table 3.

We fine-tuned each LLM using 2 DGX nodes, each containing 8 NVIDIA H100 GPUs. Our training time was an average of 7 minutes and 56 seconds. For generation, we use a temperature and top-p of 0.6 and 0.9, respectively, in order to balance both creativity and clarity in the outputs.

## 3.4 EVALUATION CRITERIA

We evaluate the results of the RAG system based on both objective and subjective criteria. We identify the following objective metrics to characterize our results:

- Exact match (EM): 1 if the answer exactly matches the target (reference) answer; 0 otherwise. We use the implementation in Rajpurkar et al. (2016), which ignores capitalization and punctuation.
- F1 score (F1): Computed at the word level as in Rajpurkar et al. (2016).
- Embedding cosine similarity (ECS): Similarity metric between embeddings of the generated and target answers.
- Metric for Evaluation of Translation with Explicit ORdering (METEOR; M in tables): Weighted F1 score between matched generated and reference answers. METEOR is somewhat robust to paraphrases or other non-exact matches (Banerjee & Lavie, 2005).
- BERTScore (B): Computes a BERT embedding for each token and scores with room for incorrect orderings (Zhang et al., 2020).

To evaluate semantic similarity, we additionally conduct experiments with an LLM-as-a-judge (Gu et al., 2025). We evaluate correctness based on the prompt presented in Listing 3. Specifically, correctness involves assessing the factual accuracy of a system's response based on the user query and the reference answer. An LLM adjudicator is provided with a grading template to assess correctness on an ordinal scale from 1 to 5. A score of 1 is assigned if the generated answer is irrelevant to the user query, scores of 2 or 3 are given if the answer is relevant but contains errors, and scores of 4 or 5 are awarded if the answer is both relevant and entirely correct. Importantly, we apply a different class of LLM, Mixtral 8x22b 0.1 Instruct (Jiang et al., 2024), for evaluation as LLMs can favor their own generations (Panickssery et al., 2024). We use a temperature of 0.6 and top-p of 0.6 for all LLM-as-a-judge evaluation.

| Retriever | Generator | EM | F1 | LJC | ECS | B | M |
|---|---|---|---|---|---|---|---|
| MPNet | Base | 0.10 | 0.21 | 2.44 | 0.36 | 0.50 | 0.20 |
| MPNet | *G-Real* | 0.28 | 0.38 | 2.65 | 0.61 | **0.73** | 0.28 |
| MPNet | *G-Synthetic* | **0.29** | **0.39** | **2.83** | **0.62** | 0.72 | **0.30** |
| GoldenRetriever | Base | 0.31 | 0.51 | 4.46 | 0.64 | 0.68 | 0.50 |
| GoldenRetriever | *G-Real* | **0.68** | **0.82** | **4.58** | **0.88** | **0.89** | 0.64 |
| GoldenRetriever | *G-Synthetic* | 0.63 | 0.78 | 4.56 | 0.86 | 0.87 | **0.68** |

Table 4: Performance of base LLM, *G-real*, and *G-synthetic* on the in-domain HotpotQA dev set. With an idealized retriever, results slightly favor *G-real*, but *G-synthetic* outperforms on almost all metrics in the presence of distractors, with the largest gain originating from LLM-judged correctness.

| Dataset | Retriever | Generator | EM | F1 | LJC | ECS | B | M |
|---|---|---|---|---|---|---|---|---|
| RepLiQA | MPNet | Base | 0.03 | 0.25 | 3.16 | 0.39 | 0.60 | 0.33 |
| RepLiQA | MPNet | *G-Real* | 0.06 | 0.24 | 2.42 | 0.42 | 0.62 | 0.19 |
| RepLiQA | MPNet | *G-Synthetic* | **0.07** | **0.33** | **3.28** | **0.52** | **0.69** | **0.34** |
| RepLiQA | GoldenRetriever | Base | 0.06 | 0.43 | **4.46** | 0.60 | 0.73 | **0.55** |
| RepLiQA | GoldenRetriever | *G-Real* | 0.10 | 0.38 | 3.19 | 0.51 | 0.67 | 0.30 |
| RepLiQA | GoldenRetriever | *G-Synthetic* | **0.13** | **0.48** | 4.07 | **0.61** | **0.74** | 0.47 |
| BioASQ | MPNet | Base | 0.00 | **0.29** | **3.47** | **0.59** | **0.60** | **0.26** |
| BioASQ | MPNet | *G-real* | 0.00 | 0.13 | 2.33 | 0.33 | 0.48 | 0.06 |
| BioASQ | MPNet | *G-synthetic* | **0.01** | 0.19 | 3.00 | 0.44 | 0.55 | 0.12 |
| BioASQ | GoldenRetriever | Base | **0.01** | **0.36** | **4.30** | **0.72** | **0.67** | **0.33** |
| BioASQ | GoldenRetriever | *G-real* | 0.01 | 0.15 | 2.66 | 0.36 | 0.50 | 0.07 |
| BioASQ | GoldenRetriever | *G-synthetic* | 0.01 | 0.24 | 3.52 | 0.48 | 0.57 | 0.16 |

Table 5: Generalization performance of base LLM, *G-real*, and *G-synthetic* on alternative (RepLiQA) or out-of-domain (BioASQ) QA tasks. 1) *G-synthetic* outperforms *G-real* across all metrics. 2) *G-synthetic* outperforms baseline significantly on all metrics for an alternative in-domain task, RepLiQA, when the retriever is imperfect and is competitive even for the GoldenRetriever.

## 4 RESULTS

We report all metrics for the baseline model along with fine-tuned versions on real and synthetic data in Table 4. We further report generalization performance to an alternate task, RepLiQA, and an out-of-domain QA task, BioASQ, in Table 5. From these results, we draw several conclusions.

**Generator fine-tuning on synthetic data improves RAG performance under an imperfect MPNet retriever.** *G-synthetic* achieved higher performance with the MPNet retriever across both HotpotQA and RepliQA and all metrics, with the exception of BERTScore for HotpotQA. Specifically, we observed an LLM-judged correctness of 2.83 on HotpotQA, a modest 6.8% increase over *G-real* and a 16% increase over the baseline. To confirm this trend, we perform a small-scale human evaluation, which may be found in Table 6. We saw a similar 3.8% increase in LLM-judged correctness for RepliQA with respect to the baseline. In mission-critical RAG systems, these increases in correctness are substantial and could be worth the cost of fine-tuning. For the GoldenRetriever, the trend is less clear. *G-real* outperformed on HotpotQA by 5.1% on F1 but only a slight 0.4% on LLM-judged correctness, likely within the margin of error. We see mixed results for RepLiQA. We explore this trend in the following sections.

| | Human-Judged Correctness |
|---|---|
| *G-real* | 2.96 |
| *G-synthetic* | 3.14 |

Table 6: Small-scale human evaluation of 100 HotpotQA dev QA pairs with the MPNet retriever. Human evaluators find a 5.7% increase in correctness for the synthetically trained LLM over that with real data.

|          | Base   | Real  | Synthetic | Reference |
|----------|--------|-------|-----------|-----------|
| HotpotQA | 50.79  | 15.67 | 18.17     | 15.43     |
| RepLiQA  | 214.88 | 58.49 | 116.99    | 80.10     |
| BioASQ   | 166.11 | 28.12 | 55.85     | 236.52    |

Table 7: Length of the generated responses compared to reference answers for each dataset. Fine-tuning on real HotpotQA data leads to responses which most closely match the length of those of HotpotQA and RepLiQA.

**Synthetically-fine-tuned LLM generalizes better to alternative QA tasks than that with real data across all metrics, regardless of retriever.** Although real data seemed to result in superior generation for the GoldenRetriever on the HotpotQA dataset, *G-real* floundered with respect to both retrievers on the alternative RepLiQA task. For MPNet, the largest differences between *G-synthetic* and *G-real* were on METEOR (-44%), F1 (-27%), and LLM-judged correctenss (-26%). Notably, these are the metrics most strongly associated with correctness besides the often too-strict exact match score, and *G-real* underperformed even the baseline model on each across both retrievers. This demonstrates a significant generalization risk in fine-tuning a generator on real data.

**Fine-tuning outperformed the baseline both due to improvements in usage of the contexts and better answer alignment to the target distribution.** For HotpotQA, we observed an average 148% increase in exact match and 70% increase in F1 post-fine-tuning. However, as shown in Figure 2, LLM-judged correctness improves by a much smaller 7.4%. Of all our metrics, EM and F1 are the most sensitive to small differences in answer style that have little bearing on semantic meaning. The fact that these metrics improved by the highest factor shows that the primary benefit of generator fine-tuning for a particular target QA task is in conformity to the desired answer style. Table 7 demonstrates how fine-tuning on HotpotQA decreases the average response length by 76%, aligning it with the short (often single word) answers of HotpotQA. Figure 4 shows how a more concise answer can lead to little increase in correctness, but a large boost in EM and F1.

The higher LLM-judged correctness of fine-tuned LLMs over the baseline indicates that answer conformity is not the only factor at play, however. Our results suggest that fine-tuning structurally improves the LLM's reasoning over passages amid distractors. A similar phenomenon is observed in other work such as Zhang et al. (2024), which shows that training over distractor documents improves the LLM's performance—a benefit that would not be observed if not for improved analysis of the contexts. Similarly, we attribute the boost in correctness partially to improved ability to leverage the documents, ignoring distractors, for the QA task.

**Fine-tuning with real data excels at conforming answers to the style of the target task, while synthetic data improves reasoning over contexts, especially amid distractor documents.** Table 7 shows that the average *G-synthetic* answer was longer than their *G-real* counterparts. This seems obvious as the verbose LLM is not predisposed to making synthetic QA pairs in the short, pointed style of HotpotQA. We would expect this to apply downward pressure to *G-synthetic* EM, F1, and LLM-judged correctness scores for in-distribution tasks as the LLM does not quite learn to generate responses in the style of HotpotQA, and we indeed see evidence that this is the case. When the LLM has all relevant contexts without distractors as with the GoldenRetriever, *G-real* or the baseline Llama-3 generator are able to outperform *G-synthetic* on some metrics. Yet in the presence of missing contexts and distractors, *G-synthetic* almost always outperforms. The impediment of not being trained on the target answer style is no match for the benefit of learning to reason over the contexts when it comes to LLM-as-a-judge correctness. Put another way, we find that the performance of fine-tuned RAG generators can depend on the retrieval quality. This is a key finding that, to our knowledge, has not been well explored in prior research.

## 5 DISCUSSION

We return to our original research question: *Can fine-tuning on synthetic data improve generalized QA performance beyond that of real training data?* We first recall why fine-tuning impacts performance at all. Some prior work has sought to use fine-tuning to embed new parametric knowledge into the LLM itself (Ratnakar et al. (2025); Liu et al. (2025a)). Others fine-tune with loss computed only over the QA answers (Liu et al., 2025b) (Zhang et al., 2024). The difference in these

approaches is stark: one attempts to increase the LLM's comprehension, while the other seeks to improve its attention over tokens in the RAG context which may improve its performance. In our approach, gradients are computed only over the answer token, but the transformer architecture includes a query and key which determine how it attends to the other tokens' values in the sequence (Vaswani et al., 2017). In other words, although we are not training the model to better generate the training passages, we are teaching it to better attend to them in the RAG context.

In some sense, the *content* of the fine-tuning QA pairs are not important, but rather their ability to teach the LLM to attend to the right elements in order to achieve the correct answer. This could partially explain why *G-synthetic* outperformed *G-real* with an imperfect retriever. With the Golden-Retriever, the LLM had all the right ingredients to give a proper answer. With an imperfect retriever, it had to "read between the lines" to reason about the right response. The authors of HotpotQA classified their turked questions into three categories: easy, medium, and hard; the dev set contains only hard questions, while the train set has all three. Supported by our qualitative and quantitative analysis of both datasets, we hypothesize the synthetically generated questions were more challenging and grounded in the contexts, leading the LLM better reason over the multiple passages, ignoring irrelevant or "distracting" information. In other words, our working hypothesis is that *synthetic data facilitates LLMs that are more robust to missing or distractor documents due to its increased ability to have the LLM rely on what is clearly stated in the passage itself.*

This difference in performance presents both a risk and an opportunity. Since performance is highly dependent on the synthetic data distribution, one change to our generation prompt could have led to drastically different results. Who knows? If our synthetic answers had been more concise in the style of HotpotQA, *G-synthetic* may have outperformed *G-real* under both retrievers. But the risk also offers an opportunity to characterize and optimize the synthetic data for the particular retriever accuracy and task.

One motif of enduring interest is the performance (and in some cases, outperformance) of synthetic data. Assume a baseline LLM generates text of a distribution $\mathcal{D}$. Our goal is to have the LLM produce data from the distribution of HotpotQA answers $\mathcal{D}^*$. Generation tasks are a foundational part of unsupervised learning because, if one can generate data from a particular distribution, to a degree one understands the distribution itself. One would assume that a model with the ability to generate question-answer pairs in $\mathcal{D}^*$ would have the ability to directly answer QA pairs in $\mathcal{D}^*$. But we find that generating questions approximating $\mathcal{D}^*$ and then fine-tuning on those questions creates a sort of positive feedback loop which encourages the model to better approximate $\mathcal{D}^*$ (sometimes at the expense of performance on other tasks). We see a connection to Battle & Gollapudi (2024), where an LLM was able to generate an intermediate prompt that elicited its own best response to a user query.

# 6 CONCLUSION

In this work, we study the impact of generator fine-tuning using real and synthetic data on RAG system performance. We fine-tune Llama 3.1 8B Instruct on both real and synthetically generated datasets. We apply these LLMs, along with the baseline, in a RAG context with both an MPNet retriever and our GoldenRetriever. We show fine-tuning on synthetic data improves LLM-judged correctness under an imperfect retriever and generalizes better to alternative tasks.

# 7 FUTURE WORK

We identify three avenues for future work. First, additional study could be done to characterize the generated synthetic dataset and understand its strengths and weaknesses. This could aid in understanding the "why" behind its strong performance relative to ground truth data. Secondly, additional work could be done to evaluate the RAG system, particularly with regards to trustworthiness, robustness, and hallucinations. Another potential application area could be chain-of-thought prompting and systems. If one model is better at generating new content, while our synthetically fine-tuned LLM excels at extracting information from the given context, could chain-of-thought systems benefit from multiple different LLMs with various specialties used in tandem? (Essentially, this would look like a combination of our work and Zhang et al. (2024).) Although some work has looked at the impacts of different LLMs as agents (Zeng et al., 2022), more research is needed.

### 7.0.1 ETHICS STATEMENT

We conducted a small-scale human evaluation with informed consent. No sensitive data or identifying information was collected, and participation involved minimal risk.

### 7.0.2 REPRODUCIBILITY STATEMENT

We release our synthetically generated questions based on the HotpotQA contexts, as these will depend on the inherent randomness in LLM generations. We detail fine-tuning parameters in Table 3. LLM-as-a-judge evaluations may change; our human evaluations lend credence to these scores. All other objective evaluation metrics are standardized, cited, and reproducible.

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

# A APPENDIX

## A.1 LLM USAGE

We leveraged GPT-4o for minor grammatical rewrites of the paper and to provide a simulated review of the work to identify areas for improvement. Additionally, we solicited references to other related works in addition to performing web searches for a literature review. Finally, we asked the LLM to suggest evaluation methodologies to ensure our work aligns with metrics used in the latest literature.

## A.2 RETRIEVER SETUP

We present an evaluation of the MPNet retriever by hit rate and MRR on each dataset in Table 8 and provide an algorithmic description of the GoldenRetriever in Algorithm 1. By design, the GoldenRetriever achieves perfect hit rate and MRR.

| Dataset | Hit Rate | MRR |
|---------|----------|------|
| HotpotQA | 0.73 | 0.66 |
| RepliQA | 0.41 | 0.52 |
| BioASQ | 0.62 | 0.69 |

Table 8: Hit rate and MRR for the MPNet retriever on each dataset.

---

**Algorithm 1** Golden Retriever algorithm. Returns the golden passages for any query in the provided HotpotQA train or dev set.

**Input**:

- Question $q$.
- Dictionary $\mathcal{D}$: A list of JSON entries with keys *_id*, *answer*, *question*, *supporting_facts*, and *context*.
- *context* is a list of pairs $[title, sentences]$, where $sentences$ is a list of strings.

**Output**: A list of contexts relevant to $q$, or the empty list if no relevant questions.

```
1: for d ∈ 𝒟 do
2:     Let dq = d["question"]
3:     if dq == q then
4:         Let r ← []
5:         for context ∈ d["context"] do
6:             Let text ← ""
7:             for p ∈ context[1] do
8:                 text ← text + p
9:             end for
10:            r.append(text)
11:        end for
12:        return r
13:    end if
14: end for
15: return []
```

---

## A.3 LLM OUTPUTS

Figure 3 contains sample synthetically generated question-answer pairs. Figure 4 contains sample responses from the baseline LLM and $G - synthetic$ from a real HotpotQA question.

**Example 1**
**Context 1:** The City Football Group (CFG) is the holding company established to oversee the creation and administration of a network of linked clubs and other footballing operations. The company is run as a holding company under parent company the Abu Dhabi United Group (ADUG) along with Chinese part-owners China Media Capital and CITIC Capital. The company's aim is to own a team on each continent, each with the identifier "City" in its name.
**Context 2:** City FC or just City is usually used as short-hand to refer to one of Manchester City F.C., New York City FC or Melbourne City FC, which are all association football clubs owned by the City Football Group, an organisation which bases it's identity around the cognomen "City".
**Real Question:** What holding company under the parent company Abu Dhabi United Group bases it's identity around the cognomen "city"?
**Synthetic Question:** Which organization owns Manchester City F.C., New York City FC, and Melbourne City FC?

**Example 2**
**Context 1:** Cognizant is an American multinational corporation that provides IT services, including digital, technology, consulting, and operations services. It is headquartered in Teaneck, New Jersey, United States. Cognizant is listed in the NASDAQ-100 and the S&P 500 indices. It was founded as an in-house technology unit of Dun & Bradstreet in 1994, and started serving external clients in 1996.
**Context 2:** Betsy Atkins (born 1953) is an American business executive and entrepreneur. She was an early investor in Yahoo and eBay in association with the venture capital firm, Baja LLC. She was the Former Chairman and Chief Executive Officer (CEO) of Clear Standards, Inc, a leading provider of SaaS Software enterprise carbon management and sustainability solutions. In 2010, Clear Standards was acquired by SAP. In addition she is President and Chief Executive Officer (CEO) of Baja Corp, a venture capital investment firm, which she founded in 1993. Atkins is on the Board of Directors of Cognizant, HD Supply, SL Green Realty Corp, Schneider Electric and Volvo Car Corporation. She served as Chairman of the SAP AG Advisory Board and is a member of the ZocDoc Advisory Board. She was a member of the NASDAQ LLC Exchange Board of Directors and is a member of Florida International University's Health Care Network Board of Directors. Atkins is a member of the Council on Foreign Relations.
**Real question:** What Teaneck, NJ headquartered the multinational corporation featuring Betsy Atkins on the Board of Directors?
**Synthetic question:** Which company that Betsy Atkins is on the Board of Directors for is listed in the NASDAQ-100 and the S&P 500 indices?

**Example 3**
**Context 1:** Alban Maria Johannes Berg ( ; ] ; February 9, 1885 – December 24, 1935) was an Austrian composer of the Second Viennese School. His compositional style combined Romantic lyricism with twelve-tone technique.
**Context 2:** Martin Greif, born Friedrich Hermann Frey (18 June 1839 – 1 April 1911) was a German freelance writer of poems and of dramas which were performed at the Burgtheater in Vienna and the Bavarian Court Theatre in Munich. His songs inspired compositions by Max Reger and Alban Berg, among others.
**Real question:** Which Austrian composer born February 9, 1885 was inspired by Martin Greif?
**Synthetic question:** Which German writer's songs inspired compositions by an Austrian composer of the Second Viennese School?

**Example 4**
**Context 1:** Adam Nergal Darski (born Adam Michał Darski; 10 June 1977 in Gdynia) is a Polish musician and television personality, best known for being the frontman for the black/death metal band Behemoth.
**Context 2:** Robert Conrad "Robb" Flynn (born Lawrence Matthew Cardine; July 19, 1967) is the lead vocalist and guitarist for the heavy metal band Machine Head. Flynn formed the band along with Adam Duce, Logan Mader and Tony Costanza after leaving Bay Area thrash band Vio-Lence.
**Real question:** Who was born first, Adam Darski or Robb Flynn?
**Synthetic question:** Which band is fronted by the Polish musician Adam Nergal Darski and which band is fronted by Robert Conrad "Robb" Flynn?

**Example 5**
**Context 1:** Daniel Constantine Marino Jr. (born September 15, 1961) is a former American football player who was a quarterback for the Miami Dolphins of the National Football League (NFL). The last quarterback of the quarterback class of 1983 to be taken in the first round, Marino held or currently holds dozens of NFL records associated with the quarterback position. Despite never being on a Super Bowl-winning team, he is recognized as one of the greatest quarterbacks in American football history. Best remembered for his quick release and powerful arm, Marino led the Dolphins to the playoffs ten times in his seventeen-season career. He was inducted into the Pro Football Hall of Fame in 2005 and the College Football Hall of Fame in 2002.
**Context 2:** The 1983 season was the 18th season in football for the Miami Dolphins and they sought to return to the Super Bowl after losing to the Washington Redskins in Super Bowl XVII. It was also a turning point in the team's history, as in the 1983 NFL Draft a young quarterback slipped to deep in the opening round, being passed over by such teams as division rivals New York who drafted Ken O'Brien and New England who drafted Tony Eason. With the 27th pick, the Dolphins decided to take a chance on Dan Marino. In the draft's eighth round the Dolphins also selected receiver Mark Clayton.
**Real question:** What team member of the 1983 Miami Dolphins is recognized as one of the greatest quarterbacks in American football history?
**Synthetic question:** Which quarterback, drafted by the Miami Dolphins in the first round of the 1983 NFL Draft, is recognized as one of the greatest quarterbacks in American football history?

Figure 3: Five randomly sampled context pairs and their associated real and synthetic questions. We make the following observations, labeled by example: 1) The synthetic question is not multi-hop as the question is answerable entirely from Context 2. 2) The real question contains a grammar error. 3) The synthetic question is greater in complexity, not mentioning any names and instead requiring linkage by the 'Second Viennese School.' 4) The LLM could not generate a good multi-hop question, instead asking separately about both passages. 5) In this case, the real question is not multi-hop; it is entirely answerable by Context 1.

> **Question:** What airline holding company located in Solna Municipality, Sweden owns a
> technical aircraft maintenance company located in Arlanda, Sweden?
> **Reference answer:** SAS Group
> **Baseline** SAS Group, an airline holding company located in Solna Municipality, Sweden,
> owns a technical aircraft maintenance company called SAS Technical Services.
> *G-synthetic* SAS Group

Figure 4: Baseline and *G-synthetic* LLM responses to HotpotQA question. Fine-tuning improves
adherence to the target answer distribution, leading to a large jump in F1 (from 0.19 to 1.0) but no
change in LLM-judged correctness (both 5).

## A.4 LLM PROMPTS

Here we provide our LLM prompts:

1. Listing 1 provides the prompt for the RAG system.
2. Listing 2 contains the prompt for synthetic data generation.
3. Listing 3 is the prompt for the LLM-as-a-judge model.

Listing 1: RAG system prompt. `documents` and `query` refer to the retrieved document contexts
and user input question, respectively. Each prompt additionally has the following system instruction:
"You are a helpful assistant".

```
"""
Context information is below.
---------------------
{documents}
---------------------
Given the context information and not prior knowledge, answer the
    query concisely.
Query: {query}
You are a helpful assistant. Answer the question directly.
If the context does not contain the answer, provide \"no answer\"
    as the response.
Answer:
"""
```

Listing 2: Synthetic Data Generation Prompt

```
"""
You are provided a set of multiple contexts. Your task is to
    generate {num}multi-hop question(s) that requires synthesizing
     information from all the provided contexts to in order to
    answer the question. DO NOT ask multiple subquestions with a
    question. DO NOT ask any questions that can be fully answered
    by a subset of the provided contexts. Additionally, provide an
     answer to the multi-hop generated question. Follow the
    example of the examples below. Each of the examples generate {
    num}questions-answer pair(s) that require synthesizing
    information from both contexts.
(Example1) Generating 1 question-answer pair.
Contexts:
Context 1: Tom Brady (born August 3, 1977) is an American former
    football quarterback who played in the National Football
    League (NFL) for 23 seasons.
Context 2: Joseph Clifford Montana Jr. (born June 11, 1956) is an
    American former football quarterback who played in the
    National Football League (NFL) for 16 seasons, primarily with
    the San Francisco 49ers.
```

```
Output:::
Multi-hop Question: Tom Brady and Joseph Clifford Montana Jr are
    both former quarterbacks of what league?
Answer: NFL
Multi-hop Question: Who has born first, Tom Brady or Joseph
    Clifford Montana Jr?
Answer: Joseph Clifford Montana Jr.
(Example2) Generating 2 question-answer pairs.
Contexts:
Context 1: Dunkirk is a 2017 historical war thriller film written,
    directed and produced by Christopher Nolan that depicts the
    Dunkirk evacuation of World War II from the perspectives of
    the land, sea and air. Midway is a 2019 war film about the
    Battle of Midway,
Context 2: Interstellar is a 2014 epic science fiction film co-
    written, directed, and produced by Christopher Nolan.
Output:::
Multi-hop Question: What is the title of the war film directed by
    the director of Interstellar?
Answer: Dunkirk
Multi-hop Question: Which film was directed by Christopher Nolan
    released first, Dunkirk or Interstellar?
Answer: Interstellar
(Example3) Generating 3 question-answer pairs.
Contexts:
Context 1: Saigon, Vietnam located in SE Asia is known for its
    vibrant street food culture, which includes noodles dishes
    like Pho.
Context 2: The Hokkien Chinese community in Singapore, located in
    SE Asia, has a rich culinary heritage, featuring noodle dishes
    such as Hokkien Mee.
Output:::
Multi-hop Question: Pho and Hokkien Mee are both what kind of dish
    ?
Answer: Noodle dish
Multi-hop Question: Are Saigon and Singapore both in SE Asia?
Answer: Yes
Multi-hop Question: What two SE Asian countries are known for its
    food culture?
Answer: Vietnam and Singapore
(Question1) Generating {num} question-answer pair(s).
Contexts: {context}
Output:::
"""
```

Listing 3: Prompt to identify correctness in LLM-as-a-judge tasks.

```
"""
You are an expert evaluation system for a question answering
    chatbot.
You are given the following information:
- a user query, and
- a reference answer, and
- a generated answer
Your job is to judge the correctness of the generated answer using
    the reference answer. Output a single score that represents a
    holistic evaluation. You must return your response in a line
    with only the score. Do not return answers in any other format
    . On a separate line provide your reasoning for the score as
    well.
```

```
Follow these guidelines for correctness scoring:
- Your score has to be between 1 and 5, where 1 is the worst and 5
    is the best.
- If the generated answer is not relevant to the user query, you
    should give a score of 1.
- If the generated answer is relevant but contains mistakes, you
    should give a score between 2 and 3.
- If the generated answer is relevant and fully correct, you
    should give a score between 4 and 5.
Example Response:
4.0
The generated answer has the exact same metrics as the reference
    answer, but it is not as concise.
## Query {query}
## Reference answer {reference_answer}
## Generated answer {generated_answer}
Correctness score:
"""
```

