# OpenReview forum: "Improving RAG Question Answering Generation with Synthetic Data"
_ICLR.cc/2026/Conference — Submitted to ICLR 2026_

### Official Review · Reviewer_wKA9 · 2025-10-24

**Soundness:** 2
**Presentation:** 1
**Contribution:** 2
**Rating:** 2
**Confidence:** 4

**Summary:**

This paper focuses on the core issue of "limited fine-tuning of the generator due to the scarcity of real annotated data" in the RAG system. Through the technical route of GPT-4o generating HotpotQA synthetic dataset, fine-tuning Llama-3 generator, and multi-dimensional evaluation, the system verifies the improvement effect of synthetic data on the performance of the RAG generator.

**Strengths:**

This article clarifies the correlation between the quality of retrievers and fine-tuning data distribution - in the MPNet retriever scenario (with interfering documents), G-Synthetics performs better due to its stronger contextual reasoning ability (ignoring interfering information); In the GoldenRetriever scenario, G-real has a slight advantage due to its short answer style that is more in line with HotpotQA.

**Weaknesses:**

1. The core differences between this study and related work have not been thoroughly compared. It is recommended to supplement quantitative comparisons with similar synthetic data studies (such as data generation costs and overall performance of the RAG system after fine-tuning) to clarify the technical increment of this study.

2. Only using HotpotQA (multi hop open domain QA) as the training dataset, without extending it to other typical RAG scenarios (such as domain specific QA, long document QA, conversational QA), cannot verify the applicability of synthesized data in different tasks; The model used is only Llama-3.1 8B Instruction, and larger scale models have not been tested.

3. The illusion rate of the model after fine-tuning has not been evaluated (such as whether the generated answers have support in the retrieval context), and one of the core requirements of the RAG system is to "reduce illusions"; Only by using LLM to indirectly measure the accuracy of facts, but without comparing the differences in "illusion types" (such as creating something out of nothing and misreading context) between the fine-tuning models of real and synthetic data, it is difficult to comprehensively evaluate the impact of synthetic data on the quality of generation.

**Questions:**

see above

---

### Official Review · Reviewer_QR7X · 2025-11-01

**Soundness:** 2
**Presentation:** 2
**Contribution:** 1
**Rating:** 2
**Confidence:** 5

**Summary:**

This paper aims to address the scarcity of high-quality real-world QA data for RAG generator fine-tuning by exploring synthetic data as a substitute. It uses GPT-4o to generate synthetic QA pairs based on the HotpotQA dataset’s context, then fine-tunes a Llama-3 generator with both real (G-real) and synthetic (G-synthetic) data. Performance is evaluated via metrics like F1, Bertscore, and LLM-as-a-judge, with additional tests on RepLiQA (same-domain) and BioASQ (cross-domain) for generalization. The paper claims synthetic data can replace real data (especially under imperfect retrievers like MPNet) and offers stronger generalization, framing this as its core contribution.

**Strengths:**

The experimental design maintains basic controllability for its core comparison: by using the same base model (Llama-3.1 8B Instruct) and fine-tuning pipeline (SFT) for both real and synthetic data, it isolates "data type" as the key variable, which allows for direct observation of performance differences between the two data sources. This basic rigor ensures the comparison itself is internally consistent.

**Weaknesses:**

- **No Substantive Innovation**: Using synthetic data for model fine-tuning is already a common practice in industry, and the paper does not propose any new mechanism for synthetic data generation, fine-tuning optimization, or performance improvement. It merely repeats existing workflows without advancing the field.
- **Extremely Limited Experimental Scope Undermines Conclusion Validity**: Training and core validation rely solely on the HotpotQA dataset—a single benchmark that has long been a standard testbed for multi-hop QA. For generalization testing, it only includes one additional cross-domain dataset (BioASQ) and one same-domain dataset (RepLiQA), far fewer than comprehensive RAG evaluations that cover multiple domains (e.g., CRAG’s five domains). This narrow scope makes it impossible to prove the "generality" of synthetic data claimed in the paper.
- **Critical Evaluations Missing**: It ignores key RAG performance dimensions that industry and academia prioritize, such as hallucination (a core risk for real-world deployment, evaluated in frameworks like RAGEval), faithfulness to context (a focus of RAGAS), and efficiency tradeoffs. This makes the work irrelevant to practical deployment needs.
- **Overstated Title and Weak Conclusion Support**: The title "Improving RAG Question Answering Generation with Synthetic Data" implies a meaningful advancement, but the limited experiments (single training dataset, sparse generalization tests) cannot support claims of "improving" RAG systems—let alone establishing synthetic data as a reliable solution.
- **Neglect of Key Optimization Directions**: It does not discuss or test the role of RL in enhancing generalization—an approach widely explored for aligning LLM outputs with real-world needs—further limiting the work’s depth.

**Questions:**

1. Given that synthetic data is already widely used in industry, what specific value does this work provide beyond confirming a known practice? How does it address gaps in existing industrial workflows?
2. The paper claims synthetic data has "strong generalization," but only tests two additional datasets. How would you design experiments to truly verify generality—for example, across more diverse domains (e.g., finance, sports as in CRAG) or more complex question types (e.g., counterfactual queries)?
3. Why did you omit evaluations of hallucination and context faithfulness—criteria central to RAG reliability? Do you have preliminary data on whether synthetic data fine-tuning exacerbates or mitigates hallucinations?
4. The title emphasizes "improving" RAG performance, but your results are limited to a single base model and narrow datasets. How do you justify this framing given the lack of evidence for broad applicability?

---

### Official Review · Reviewer_JVib · 2025-11-01

**Soundness:** 2
**Presentation:** 3
**Contribution:** 1
**Rating:** 2
**Confidence:** 4

**Summary:**

This paper compares synthetic datasets with real QA data in RAG settings. The key question is whether synthetic QA data can effectively replace or complement real QA data. Using HotpotQA as the seed dataset, the authors generate a synthetic version of it using GPT-4o, fine-tune separate Llama-3.1-8B models on the real and synthetic data, and evaluate their performance under two retrieval settings: an imperfect MPNet retriever and a GoldenRetriever.

Results show that fine-tuning on synthetic data can match or even exceed fine-tuning on real data, particularly when retrieval is noisy or incomplete. The synthetically trained model generalizes better to new datasets such as RepLiQA and exhibits greater robustness under retrieval imperfections.

Overall, the paper provides an empirical analysis of generator fine-tuning under data and retrieval constraints with practical insights for deploying RAG systems in low-data settings.

**Strengths:**

1. The paper is well written, logically structured, and easy to follow.

2. At a high level, the study addresses an important and realistic scenario of how to improve RAG performance when real fine-tuning data is scarce, sensitive, or unavailable.

3. The analysis of how retriever quality interacts with fine-tuning data quality is a useful and underexplored systems-level perspective.

4. The results provide two clear empirical conclusions:

4.1 Synthetic fine-tuning helps the generator reason more effectively under noisy retrieval.

4.2 Real fine-tuning mainly improves answer style conformity (shorter, HotpotQA-like answers).

**Weaknesses:**

1. Limited experiments: The entire analysis is centered on HotPotQA (which requires at most 2 sub-queries). Although the authors test transfer on RepLiQA and BioASQ, both are relatively small and do not fully validate generalization across reasoning types or domains. The paper makes broad claims about synthetic data but only use a single seed set which does not provide enough evidence.

2.  The paper does not compare against state-of-the-art fine-tuning frameworks such as Self-RAG, RQRAG, LeReT, CoRAG, or RAFT. These systems include retrieval-aware or reflection-based fine-tuning that may addresses many of the issues of imperfect retrieval. Without these comparisons, the reported gains are difficult to interpret broadly and have limited practical utility.

3. The paper’s novelty is primarily empirical. Synthetic data generation, RAG fine-tuning, and LLM-as-a-judge evaluations are all established techniques. The new contribution lies in the analysis, and the conceptual and methodological novelty is limited.

4. The paper quantifies embedding similarity but does not probe *why* synthetic data leads to better generalization. For example, do synthetic examples exhibit more reasoning steps, cleaner style, or reduced ambiguity? A deeper analysis would be beneficial to the community.

5. The analysis done in the paper is very limited: the effect of synthetic data size, diversity, prompt construction, etc. is not explored which might play a significant role in the overall results.

6. In relation to prior work: the paper mentions how their proposed analysis is different from methods like RAG-Studio, RAFT, etc. in the related works. However, it does not do justice to how similar some of these methods are to the analysis presented in the proposed work. For example RAG-Studio also uses synthetic data for domain adaptation and provides several ablations that reveal similar findings. I would request the authors to do a more comprehensive comparison with related works to establish their positioning.

7. As stated in the previous point, while the authors claim novelty in analyzing performance under varying retriever quality, earlier works such as Balaguer et al. (2024) and Siriwardhana et al. (2023) already explored end-to-end retriever–generator tradeoffs. The added insight here is incremental.

**Questions:**

Kindly see weaknesses.

---

### Official Review · Reviewer_bDWW · 2025-11-02

**Soundness:** 2
**Presentation:** 2
**Contribution:** 1
**Rating:** 2
**Confidence:** 4

**Summary:**

This paper studies whether the synthetic data can substitute the golden data when fine-tuning the generator in a RAG system. GPT-4o is applied to generated the synthetic HotpotQA like questions based on the same golden passage pairs as the HoptpotQA training set. Llama-3.1-8B-Instruct is selected as the generator and trained on golden training set and the synthetic training set separately. Metrics, like EM, F1, BERT Score and LLM-as-a-judge are utilized to compare the performances. The author concludes that the generator fine-tuned on synthetic data outperforms the counterpart fine-tuned on golden training set with the imperfect retriever, MPNet. In addition, the generator fine-tuned on the synthetic data has better generalization than its counter-part.

**Strengths:**

1. The author clearly describes the experimental setup, and the Figure 1 task diagram is helpful to understand the baselines.
2. The synthetic QA dataset is released and may be helpful for the research in this area

**Weaknesses:**

1. The synthetic questions are generated from the same golden passages as real HotpotQA, making the claims overestimated.
2. The synthetic data generation and fine-tuning are only conducted on HotpotQA, raising the concerns about the generalization.
3. The usefulness of synthetic data has been proved in many previous studies, and it is not supervise to observe the benefits of using the synthetic data to train the generator.

**Questions:**

The primary challenge in generating multi-hop questions lies in collecting the linked passages required to answer them. This paper bypasses that difficulty by directly using the same passage pairs as the HotpotQA dataset. Why not consider a more general synthetic data generation method?

---

### Meta-Review · Area_Chair_VoXe · 2026-01-07

**Summary:**

The paper explores the utility of synthetic data for fine-tuning the generator in a RAG system. While it provides empirical evidence that synthetic data can serve as a substitute for real data, the reviewers raised concerns regarding the paper’s lack of methodological novelty and the limited experimental scope. The AC agrees with the consensus that the work primarily confirms established industrial practices rather than offering new technical contributions or a comprehensive comparison with existing frameworks.

**Reviewer Concerns:**

### Addressed by Rebuttal
None. The authors did not provide a rebuttal or participate in the discussion.

### Still Outstanding
The primary concerns remain regarding the lack of method novelty and technical contribution, as the approach largely confirms established practices without advancing the field. The experimental scope is narrow, relying on a single dataset and omitting critical RAG evaluations such as hallucination rates and context faithfulness. Furthermore, reliance on identical passages for synthetic data generation raises significant concerns about the potential overestimation of the model's performance.

**Reviewer Scores:**

Since no rebuttal was provided to address the weaknesses, the AC believes the reviewers' scores would have likely remained unchanged even if a full discussion period has taken place.

---

### Decision · Program_Chairs · 2026-01-26

Reject